



# Soil carbon estimates by Yasso15 model improved with state data assimilation

Toni Viskari[1], Maisa Laine[1], Liisa Kulmala[1,2,3], Jarmo Mäkelä[1], Istem Fer[1] and Jari Liski[1]

[1]Finnish Meteorological Institute, Helsinki, 00101, Finland
[2]Department of Forest Sciences, University of Helsinki, P.O. Box 27, FI-00014 Helsinki, Finland.
[3]Institute for Atmospheric Sciences and Earth System Research, University of Helsinki, Helsinki, Finland

*Correspondence to*: Toni Viskari (toni.viskari@fmi.fi)

**Abstract** Model-calculated forecasts of soil organic carbon (SOC) are important for approximating global terrestrial carbon pools and assessing their change. However, the lack of detailed observations limits the reliability and applicability of these SOC projections. Here, we studied if state data assimilation (SDA) can be used to continuously update the modeled state with available total carbon measurements in order to improve future SOC estimations. We chose six fallow test sites with measurements time series spanning 30 to 80 years for this initial test. In all cases, SDA improved future projections but to varying degrees. Furthermore, already including the first few measurements impacted the state enough to reduce the error in decades long projections in by at least 1 t C ha$^{-1}$. Our results show the benefits of implementing SDA methods for forecasting SOC, but also highlight implementation aspects that need consideration and further research.

## 1. Introduction

Terrestrial soil organic carbon (SOC) pools serve a crucial role in the global carbon cycle by acting as a large long-term carbon storage for terrestrial systems and are, similarly to the other carbon cycle components, directly impacted by the changing climate and environment (Ciais et al., 2013). Local meteorological conditions drive soil temperature and moisture, which together with soil characteristics in turn affect the microbial processes that decompose SOC (Orchard and Cook, 1983; Karhu et al, 2014; Vogel et al. 2015). SOC input, on the other hand, is largely composed of surface vegetation litter and extracts with contributions from soil bacteria and mycorrhiza (Cornwell et al., 2008). Thus, when the vegetation cover is altered either due to changing environmental conditions or anthropogenic activities, it will also alter the long term SOC stocks. Furthermore, the SOC response to the new surface conditions is slow and it will take years to decades, or even longer with drastic changes such as peatland draining or transforming forests into agricultural fields, before it fully reaches a new stable state (Mao et al, 2019). All these factors have made it difficult to empirically assess how both local and global SOC stocks will be affected by the changing climate and environment (Sulman et al., 2018).



To address these challenges, several SOC models of varying complexity have been created over the years (e.g. CENTURY (Parton, 1996), MILLENIAL (Abramoff et al., 2017) and ORCHIDEE-SOM (Cammino-Serrano et al., 2018)) with an increasing focus on how to better mathematically formulate the central physical soil processes (Liang et al. 2017). These models allow projecting SOC in different environments and, thus, are important tools in approximating regional and global SOC distributions as well as how they are changing over time (Manzoni and Porporato, 2009). As such, they also serve an important role in estimating how climate change impacts the SOC stocks, which due to the size of pools and their direct link to ecosystem response is one of the largest uncertainties in future carbon cycle projections (Hararuk et al. 2014). On a practical level, SOC models have been used to calculate soil carbon components for National Carbon Budgets or determine carbon allocation in soils under different agricultural management conditions when calculating carbon credit market values (Smith et al., 2020).

Yet despite this increasing number and variety of modelling choices, the future projections produced by them all face similar difficulties which have resulted in high uncertainties in future projections (Bradford et al, 2016). Fundamental among these challenges is the lack of observation data required to parameterize and initialize the models (Sulman et al, 2018). Relevant measurement campaigns are resource-heavy and time-costly whereas SOC varies highly spatially (Jandl et al., 2014). Furthermore, vast majority of the available measurements represent bulk total soil carbon contents whereas the decomposition dynamics are greatly dependent on a more nuanced representation of the organic carbon state such as which fraction of SOC is contained by stable long-lived carbon compounds as opposed to active short-lived carbon compounds (Lehmann and Kleber, 2015). The missing detailed measurements forces models to use other, less reliable methods to approximate the initial SOC state, which in turn is a major limitation in trying to estimate how the projected SOC state reacts to environmental changes (Wutzler and Reichstein, 2007; Palosuo et al., 2012).

Using observations to constrain state projections is a central question for all predictive tasks, and different approaches have been developed to address this need. State data assimilation (SDA) refers to Bayesian methods where state information from two or more sources are combined to create a more accurate estimation of the true state (Evensen, 2009). It has already been applied in several geophysical subjects (e.g., Elbern et al. 2000; Weaver et al. 2003; Viskari et al., 2012; Yang et al., 2019) and is a fundamental component that allows weather forecasts (Le Dimet and Talagrand ,1986). In recent years there have been efforts to also use SDA methods to better incorporate flux tower and satellite (Viskari et al., 2015) measurements to update ecological model projections. One of the core advantages of SDA is that it allows to update unobserved state variables with information from observed state variables based on the currently understood and presented process dynamics (Dietze, 2017).

Applying SDA methods in SOC research could potentially address several current challenges in the field. As SDA makes it possible to continuously incorporate measurement information to update and correct the model state, it consequently both





reduces the impact of initial state uncertainty and allows using multiple measurements to better constrain future SOC projections. Due to SDA being able to update unobserved state variables based on observed ones, it allows use of the total carbon measurements to correct the more detailed model state, for example the division between active and stable SOC pools, as well as estimating regional carbon stocks based on local measurements. However, while the basic equations for SDA remain

the same, there are practical challenges in implementing SDA that depend on the system examined, such as varying frequencies for different observations or the types of observations uncertainties (Dietze, 2017). Consequently, implementing SDA for ecosystems requires addressing different issues and questions than implementing SDA in atmospheric systems (Dietze et al., 2018).

In this study, our aim was to determine if SDA efficiently improves SOC model projections using coarse observation data to continuously update the model SOC state. More specifically, we wanted to both determine how the total carbon measurements affect the individual model pools and how many measurement points need to be included for them to start impacting the future predictions in a noticeable manner. The decades long SOC dataset measured at bare fallow agricultural fields around Europe (Barré et al. 2010) was used along with Yasso (Tuomi et al., 2011; https://github.com/YASSOModel), a SOC decomposition

model that has been shown performing well for long-term SOC projections (Ortiz et al., 2013; Ziche et al., 2019), to test if updating the model projection with observations has an impact on future state predictions. The bare fallow sites do not include the uncertainty of litter input estimates and thus, allowed us to focus more on the impact SDA has on the model projections.

**2. Materials and methods**

**2.1. Yasso model**

Yasso (Tuomi et al., 2011) is a soil organic carbon (SOC) model which simulates SOC decomposition by shifting C between different soil pools representing different organic carbon forms before either releasing it back to the atmosphere as heterotrophic respiration or transforming it into inactive and slow-cycling humus. Within the model, carbon is divided into five different SOC pools: Ethanol (E), Water (W) and Acid (A) soluble pools and a non-soluble pool that is further divided in to lignin-like pool (N) and a humus (H) pool having different decomposition rates. Decomposition is affected by air

temperature and precipitation, which are used in the model as indicators for soil temperature and moisture. Additionally, Yasso accounts for the size dependency for woody mass as it takes longer in those situations for the microbes to break the litter down. Model SOC can only increase by the plant litter input.

The change in state at time $t$, $x_t^{'}$, is represented as a matrix equation

$$x_t^{'} = A\, x_t + b$$

(1.)



Where $x_t$ is a vector where each component is the amount of carbon in each pool, $b$ is the litter input and $A$ is the matrix where the diagonal values represent the fraction of mass being removed from the pool and the non-diagonal terms dictate the amount of the removed carbon transferred to other pools. The diagonal terms in matrix A are adjusted based on annual average temperature as well as temperature variation amplitude, precipitation and woody diameter. The parameters associated with each process were estimated with an Adaptive Metropolis MCMC (Haario et al., 2001) method based on joint information

from a number of different litter decomposition data bases such as CIDET (Trofymov, 1998), LIDET (Gholz et al., 2000) and Eurodeco (Berg et al, 1991a; Berg et al. 1991b).

### 2.2 The measurement time series

Bare fallow experiments included in the study were kept vegetation-free and free of organic amendments for more than 25 years. The study sites are located in Europe and selected characteristic of these are presented in Table 1. The cultivation time that lead-up to the bare fallow experiment varied from 75 years to centuries. The sites are introduced in detail by Barré *et al*. (2010).

### 2.3 State Data Assimilation method


As there is no way to know the true state of a variable, all our information on it, be it modelled or observed, will be inherently uncertain (van Oijen, 2017). State data assimilation (SDA) is a Bayesian statistical method which combines information from multiple sources, generally from model prediction and observations, to create a statistically optimal state estimate. At each assimilation step, SDA updates a priori knowledge of the system state, almost always a model prediction, with state

observations. This results in a posterior state estimate of both the expected value as well as the associated uncertainty, both of which are considered the most reliable view on the true state given the available information as the estimated posterior state uncertainty would be less than any of the sources' uncertainties. Each information source influences the posterior estimate in proportion to their uncertainties: higher observational uncertainty results in a posterior state estimate closer to the model prediction, and vice versa (Dietze, 2017).

In our research we used the Ensemble Adjustment Kalman filter, EAKF (Andersson, 2001) which is based on the Kalman Filter theory (Kalman, 1960). The ensemble consists of numerous model projections started from different initial conditions which are moved forward in time independently until the next observation and the prior state uncertainty is determined from the ensemble spread. At the time of each observation, an update (later called as analysis) is calculated with the following equation

$z_i^a = A^T(z_i^f - \bar{z}^f) + \bar{z}^a,$ (2.)



where $z$ is a joint state-observation vector, index $f$ denotes forecast, index $a$ denotes analysis and index $i$ denotes each individual ensemble member. Matrix A shifts the whole ensemble so that the updated ensemble has a mean equal to

$$\bar{z}^a = P^a[(P^f)^{-1}\bar{z}^f + H^T R^{-1} y]$$ (3.)

and covariance equal to

$$P^a = [(P^f)^{-1} + H^T R^{-1} H]^{-1},$$ (4.)

where $y$ is the observation vector, H denotes the observation operator, P is the model state error covariance matrix and R is the observation error covariance. It should be noted that the analysis error covariance matrix $P^a$ has non-diagonal terms which represent the error covariances and which allow the observation of a specific state also affect other members of the state variable vector.


There are practical challenges that need to be accounted for when utilizing SDA methods, such as an assumption of normally distributed uncertainty, difficulty in assessing model process error and filter divergence. Errors in SOC are inherently not normally distributed as SOC cannot have values below zero, but the major pools in this study are large enough that that boundary condition issue is not considered critical. With process error, even though it is assumed to be a major component in

every SDA application, there are no reliable ways to establish it for process-based simulators like Yasso yet, hence we do not account for it here. Filter divergence (Schlee et al, 1967), however, is a persistent issue that is a consequence of all the associated challenges and is the most relevant for this study. In practice, the ensemble uncertainty does not represent all the uncertainty sources affecting the model predictions and, consequently, the modelled uncertainty does not necessarily increase enough to balance out the reduction in posterior uncertainty during the analysis phase. This results in the updating process

giving too much weight to the prior state when compared to the observed state until the measurements start not affecting the estimate much, if at all, anymore, at which point the forecast begins to diverge from reality. There are several methods for dealing with filter divergence (Evensen, 2004; Anderson, 2006), but as this a preliminary study, we used a simple inflation method established in Hamill (2001), in which the forecast/prior covariance is multiplied with a constant factor greater than 1 before every analysis/update step in order for to ensure that the measurement continue to affect the estimate. The practical

implementation is explained in more detail in the following section.

### 2.4 Simulation set-up

We used the approach detailed in Kulmala and Liski (2018) to determine the site-specific initial state based on the site. First,

we made a general Net Primary Production (NPP) estimate using mean temperature and precipitation as in Del Grosso et al. (2008) and divided it into non-woody, small-sized woody and large woody litter fractions based on the native ecosystem. The





different litter types had individual carbon fractions based on solubility. Next, we used Yasso to determine the steady state pool of soil carbon and its fractions using the NPP, different litter fractions, their chemical fractions and mean temperature and precipitation as driver data.

Before bare fallow, each field had been cultivated for 75–300 years. For that period, we simulated SOC starting from the achieved steady state SOC and its chemical fractions using again the mean annual temperature and precipitation as drivers. The annual litter input for the cultivation period was estimated in a site-specific manner to meet the first SOC measurement after the cultivation period. The carbon fractions in that litter input were assumed to be as presented in Karhu et al. (2012) and the litter is assumed to be non-woody with a diameter of 0 cm. The resulting SOC as the starting point for the bare fallow

period. The AWENH distributions calculated in this manner for each site are shown in Table 1.

The ensemble initial states were created with R language using the *rnorm* function with a condition checking that the outputs are non-negative. The initial ensemble values for each pool were determined by drawing from a normal distribution where the initial value for that site was used as a mean. As there were no reliable uncertainty estimates for the initial state, a 10 % mean was used as the variance as, after testing different ways to perturb the initial state, it was decided to be wide enough for the

purposes here and was larger than just perturbing the litter inputs. The sum of each pool perturbation was the difference in total carbon. For this initial study covariances between the different SOC pools were not considered. The sampling was used to create an ensemble with 50 different states. This way we can represent the uncertainty in the total amount of soil organic carbon and how it is distributed among the five pools. The initial distributions for total carbon are shown in Fig 1.

While necessary, using the first measurement to scale the initial SOC state does raise questions regarding the SDA

implementation as it would result the first measurement to be used twice if the SDA was done over the whole time series. Not scaling the initial state would produce different results due to the large uncertainties in the prior litter input values and the resulting SDA estimations would be expected to be superior to the non-SDA predictions in that situation. In other words, it would not be a fair comparison as generally in runs like these the initial state would be constrained to some degree by available measurements. Other option could be to exclude the first observations from data assimilation. However, including the

information from the first measurement in a decomposition time series in the SDA implementation is assumed to be important as the SOC state changes most drastically over the first few years which in turn would impact the initial state uncertainty propagation. In an ideal situation there would be an independent SOC measurement that can be used to constrain the SOC initial state, but such additional data was not available here. Thus, as using the first measurement twice was expect only a very negligible effect on the overall results, we used the whole time series in the assimilation here. We also did a comparison run

where the SDA was only applied from the second measurement forward in order to be certain. These runs were set up identically except the relative error of the first measurement was used as variance to randomly draw the ensemble members.





We used the Data Assimilation Research Testbed (DART; Anderson et al, 2009) to run our assimilation with the EAKF. The initial ensemble for each site was given to DART as a starting point and climate data measured at the sites were used as model drivers. The climate driver data is provided alongside with this article. The state vector consists of the five SOC pool stocks

as presented in Yasso and the total SOC which is a sum of the five pools. The total SOC projection is compared to the measurements and the error covariances calculated by DART transfer the information to the other state vector components.

The SDA was first tested by updating the model state variables at each measurement time and using that state estimation to determine the next predicted state. This basic test was repeated with three different inflation factors (1, 1.25 and 1.5) in order

to examine how much filter divergence affects the projections and which inflation factor range produces satisfactory predictions. Only the inflation factor results for 1. and 1.25 are shown here for the sake of clarity. In the second set of tests, only a limited number of initial measurements (first, first two, first three or first four) are used to update the state before it is then allowed to run the whole time series without being updated with measurement information in order to determine how soon the measurement information begins to noticeably impact the model projections. Only the first four measurements were

used in this phase as the central question was how assimilating SOC measurements impact long term forecasts. The inflation factor of 1.25 was used in these latter tests.

All the forecasts produced with these tests were compared to both measurements and baseline Yasso SOC projection that was ran from the initial state without any SDA. In order to better assess how the SDA improved the state forecasts, we calculated

the RMSE for the last four measurements at each site using the forecasts that used the limited number of measurements as well as the baseline Yasso model forecast.

## 3. Results

Using SOC data to update the state of the model improved the model-calculated estimates compared to non-SDA model projections run from the approximated initial state (Fig. 2). While the inflated SDA predictions had larger uncertainties than the uninflated ones, the predictions themselves remained close to each other with exception of the two Askov sites. There, systematic shifts occur in the observed states decades after the start of the time series, and, indicative of the effect of the filter divergence, the unfiltered SDA predictions did not react to these shifts while the inflated SDA predictions were adjusted to the

new states. Applying the inflation value of 1.5, the analysis essentially matched the measurements (Not shown as here later in the time series the prior state is simply the previous observation).

Due to the multiple changes in the Askov B4 time series, the more detailed model state response is represented for it in order to see how the state estimate adapts to the changes there. Among the SOC pools, the humus pool changed most in response to





the observations and always to their direction at Askov B4 (Fig. 3). The SDA estimate altered AWEN pools only little during the first half of the time series, but after approximately 10 years, these pools were also changed during the state update. Interestingly, these pools were changed to the opposite direction than the humus pool and observations. SDA affected the humus pool in a same way at other sites, and a similar difference in the behaviour between the AWEN and the humus pools was observed at Askov B3 after the systematic shift (Not shown).


At most sites incorporating information already from the first two observations had a noticeable impact on the time series prediction, although at Rothamsted and especially Versailles sites, the SDA forecasts were close to the non-SDA forecasts at the end of the time series (Fig 4). At the Askov sites, the updated predictions ended up overestimating the latest measurements more than the model alone did due to the systematic shift in measurement values after 1966 at Askov B4 and after 1977 at

Askov B3. Finally, RMSE values (Table 2) show that aside from Askov, the assimilation reduced the RMSE at each site by the fourth measurement at the latest.

The comparison runs where the assimilation was only done from the second measurement forward were nearly identical for the estimated total SOC values when continuously assimilating and when only using the first few observations to constrain the

predictions (Figures not shown). The more detailed examination of the state at Askov B4 (Figure 5) did show a difference, however, where the later corrections affecting the AWEN pools are more muted than if the assimilation begun from the first measurement of the time series.

### 4. Discussion


This study establishes that state data assimilation (SDA) improves soil organic carbon (SOC) forecasts by continuously incorporating total carbon measurements. Furthermore, not only do our results show that, in almost all cases studied here, SDA produces forecasts that are closer to the future measurements than the model projection alone. At all sites assimilating already the first few measurements had a clear impact on the forecasts (Fig 4; Table 2). It should be noted that at Askov the non-SDA

forecast is closer to the measurements towards the end of the time series than the SDA forecast that assimilated the first few observations. This is due to the systematic shift in measured SOC that happens at Askov B3 around 1965 and at Askov B4 around 1975. Before that, the SDA forecasts are closer to the SOC measurements than the non-SDA forecasts. This supports previous research on the impact of initial state uncertainty on SOC projections (Todd-Brown et al. 2014; He et al. 2016).

While the inflation term does increase the uncertainty of the forecasts and thus reduces the filter divergence, the uninflated and inflated forecasts remain close to each other. Askov sites are the exception here as there the inflated SDA forecast reacts to the previously noted systematic change. These results here indicate that it succeeds in the framework discussed in Anderson (2001)





on how inflated systems should behave. However, once litter input will be introduced into the system, it will add a potentially systematic source of error as the uncertainties in the litter input affect the SOC projections. At that point a more nuanced

inflation approach or other more elaborate implementations, such as estimating the process variance from observations (Dietze, 2017), could be required.

When examining how SDA affects the model state, it is important to note that the total SOC measurements affect the model state based on the error covariances between the different pools and their total SOC. The initial uncertainties were introduced

as independent of each other with the SDA calculating the error covariances between the different pools over the analysis process. The resulting error covariance between the humus pool and total SOC is a strong positive one with a decrease in humus also decreasing total SOC and vice versa (Fig. 3). This is reasonable as the long-lived SOC is generally dominated by the stable humus pool (Lehmann and Kleber, 2015) and adjusting to it is crucial in capturing the decomposition without litter input. Furthermore, resulting to the slow decomposition rate of the H pool and the relatively high frequency of the observations,

at each assimilation time the prior H value is essentially the posterior H of the previous assimilation cycle.

Error covariances between the AWEN pools and total SOC are more complicated and thus it takes more analysis cycles for the method to establish them. Consequently, the analysis appears to affect the humus SOC from the start of the time series while with AWEN pools the analysis impact appears to become stronger later into the time series. Due to this, the two Askov

timeseries are the only sites here where we also capture the meaningful AWEN pool impacts due to the late shift in the observed state. Even there, though, that covariance is strongly affected if the uncertainty spread over the first few years of the decomposition is included (Figs 3 and S1). It is noteworthy that once the SDA properly determines the error covariance structures, the analysis adjusts the AWEN pools to the opposite direction than it does the H pool. Initially this might appear to be counterintuitive, as in response for the forecast overestimating the SOC values, SDA increases the AWEN values, but this

is due to model dynamics being reflected via the error covariances. In the case of Askov B4, not only does SDA reduce the H SOC, it also essentially shifts some of that carbon back to the AWEN pools where the heterotrophic respiration rate is higher than in the H pool. Thus, SDA tries to correct the system through model dynamics, which would assume that a reason for the difference is that the transition from the AWEN pools to the H pool has been too fast. Further complicating the matter is that the active AWEN pools are affected differently by the environmental conditions than the inactive H pools (Tuomi et al, 2008)

which will cause the model dynamics and consequently the resulting AWENH error covariances to vary between locations even if the total carbon and H error covariance appears to be consistent.

In addition to providing a valuable illustration in how the error covariances change over time and impact the later state corrections, the Askov sites also highlight both a particular strength and limitation of the SDA methods. As seen in the

measurement time series (Fig 2), both Askov B3 and B4 have a systematic shift in measurements at different times due to reasons currently not known. In both cases, the inflated SDA adapts to the new state within a few measurement cycles and





produces a forecast that follows the new state well. This is clearly a strength of the SDA method that would be beneficial when forecasting SOC at locations where there are disturbances and alterations in the surface conditions. However, here when adjusting to the new state within the context of the prior information, SDA creates a new state estimate that appears

questionable as there is a sudden increase of SOC in active pools despite it being over a decade since there was any litter fall. Thus, while SDA is a beneficial tool when examining changing systems, the nature of SOC model dynamics makes it important to also expertly assess how the new estimated state reacts to those changes.

Another site that shows SDA having challenges is Versailles where SDA only slightly improves the forecasts towards the end

of the time series. This is probably due to either the model not representing a dynamic affecting SOC decomposition at that site or the input drivers being lacking in some manner. This site highlights that while SDA is a valuable tool for improving forecasts, it is still limited by how well the applied model captures the local SOC dynamics. However, SDA is still useful in these situations as it can indicate sites where the forecast error is not driven by the state uncertainty and thus make it easier to analyse the differences between the sites like Ultuna and Versailles. For example, it is known that soil quality affects the SOC

decomposition (Chapin et al. 2011; Vogel et al. 2015), so here it would support in further researching the soil properties at Versailles to determine if those dynamics have an impact there that should be acknowledged with SOC forecasts at other similar sites.

The continuous SDA forecasts from Askov sites and Versailles (Fig. 2) also indicate the complexity of the filter divergence

issue in SOC systems and how it should be accounted for. As explained in section 2.2, one of the key reasons for filter divergence is due to the prior state uncertainty being underestimated due to ignoring of model process error which results in the prior state being given progressively more and more weight in the assimilation phase. At more frequently measured sites, such as Askov B3 and B4, there are more assimilation steps, which would intuitively speed up the filter divergence issue. However, as can be seen in Eq. 3., the reduction in posterior uncertainty depends on the observation uncertainty, with less

uncertain observations also reducing the posterior uncertainty more. Thus, at the Askov site, the measurement uncertainties are large enough that it partially balances out the measurement frequency and the resulting forecast uncertainty is large enough to allow for rapid adaptation to changes in the system.

At Versailles, though, while the measurements are much less frequent, they also have small associated uncertainties, especially

the first few ones. Furthermore, for long decomposition systems like this, the uncertainty propagation within the ensemble is so slow that it only marginally increases the state uncertainty until the next observation point, resulting in filter divergence becoming an even more pronounced issue here. As a result, even with uncertainty inflation, the first few assimilation steps reduce the state uncertainties to the degree that the difference between projections and measurements affects the state estimate much less than at the other sites. The new observations still affected the inflated SDA, as can be seen at the last Versailles

measurement in Fig 2, but it will take multiple observations with increasing difference between forecasted and measured state





for SDA to properly adjust to the new state. This highlights the importance to carefully consider the relationships between the observation uncertainty, frequency and inflation in order to improve the assimilation results. This is also a general issue within the application of SDA in geosciences and, as such, there have already been attempts to mathematically address it such as Li et al. (2009).


## 5. Conclusion

The results here show that there are benefits in implementing SDA methods in SOC research and projections, but also highlights the need for additional study. The focus here was in a very simply system where there was no litter input and on a

specific SDA method with its own benefits and hindrances. Increasing the complexity of the system, such as by introducing different types of litter, using measurements from other locations to estimate local SOC or incorporating flux tower respiration measurements to constrain projected SOC changes, also raises new practical challenges as well as enhances those noted here as the litter input will affect the AWEN pools in a steady state and thus rises the importance of correctly representing the associated error covariances from the start. Still, by allowing actively incorporating multiple information sources, SDA is a

crucial tool for all process-based model projections, be it from approximating the amount of SOC in a forest to assessing how agricultural carbon allocation changes in response to field management.

### Acknowledgments

Academy of Finland (grant numbers 297350, 277623) and ERA-NET FACCE ERA-GAS project FORCLIMIT are

acknowledged for financial support. FACCE ERA-GAS has received funding from the European Union's Horizon 2020 research and innovation programme under grant agreement No 696356.

### Author contributions

Dr. Toni Viskari came up with the study, planned the experiments and wrote majority of the manuscript. M.Sc. Maisa Laine

did the simulations and the initial analysis of the results. Doc. Liisa Kulmala provided the data and expertise on the measurement time series used in the study. Drs. Jarmo Mäkelä and Istem Fer provided mathematical insight into the methods used as well as the interpretation of the results. Prof. Jari Liski is the PI of the project this research was a part of and has created the Yasso model used here.

**Data availability**

The Yasso model and the parameters used here can be downloaded from https://github.com/YASSOmodel/YASSO15. The data and scripts used to run the model and produce the figures can be accesses at https://github.com/Viskari/Yasso_SDA_Data. The permanent version of the Yasso15 code and data used in this publication has also been uploaded to Zenodo (doi:10.5281/zenodo.3891133).





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





| Site | Askov (B3, B4) | Grignon | Rothamsted | Ultuna | Versailles |
|---|---|---|---|---|---|
| Country | DE | FR | UK | SWE | FR |
| Mean temperature (°C) | 7.8 | 10.7 | 9.5 | 5.5 | 10.7 |
| Precipitation (mm) | 862 | 649 | 712 | 533 | 628 |
| History | arable | Arable | grassland | grassland | grassland |
| Bare fallow starting | 1956 | 1959 | 1959 | 1956 | 1928 |
| Soil type (FAO) | Luvisol | Luvisol | Luvisol | Cambisol | Luvisol |
| Clay-silt-sand (%) | 7-11-82 | 30-54-16 | 25-62-13 | 36-41-23 | 17-57-26 |
| Bulk density (kg dm$^{-3}$) | 1.5 | 1.2 | 0.94 | 1.44 | 1.3 |
| Fertilization | ✓ | – | – | – | – |
| Tillage frequency | frequent | 2/year | 2-4/year | 1/year | 2/year |
| Weeding by hand | ✓ | ✓ | – | ✓ | ✓ |
| Measurement time series | 1956-1985 | 1959-2007 | 1959-2008 | 1956-2007 | 1929-2008 |
| Initial AWENH carbon pools (tC/ha) | (4.4,0.5,0.3,8.9,38.1) (3.4,0.3,0.2,6.9,36.9) | 4.7,0.5,0.3,11.3,25.0 | 10.7,1.1,0.6,23.5,35.8 | 5.9,0.6,0.4,12.9,22.7 | 8.8,0.9,0.5,20.8,34.5 |

**Table 1**: The bare fallow sites used in this study. The Askov site was fertilized by 70 kg N/ha until 1973 and by 100 kg after

that. Before bare fallow, Askov was cultivated since 1800, Grignon since 1875, Kursk since app. 1765, and Versailles since

17$^{th}$ century. Ultuna has been experimental field for agriculture for centuries. There are two different plots at Askov site (B3

and B4) with different initial state values.





| | Non-SDA Yasso | | First measurement assimilated | | First two measurements assimilated | | First three measurements assimilated | | First four measurements assimilated | |
|---|---|---|---|---|---|---|---|---|---|---|
| | RMSE (t C ha$^{-1}$) | MRE (%) | RMSE (t C ha$^{-1}$) | MRE (%) | RMSE (t C ha$^{-1}$) | MRE (%) | RMSE (t C ha$^{-1}$) | MRE (%) | RMSE (t C ha$^{-1}$) | MRE (%) |
| Askov B3 | 2.5 | 7.0 | 2.6 | 7.3 | 2.5 | 7.0 | 3.3 | 9.4 | 3.5 | 9.8 |
| Askov B4 | 4.0 | 12 | 4.1 | 12 | 5.8 | 17 | 6.0 | 18 | 5.9 | 18 |
| Grignon | 2.6 | 8.6 | 2.8 | 9.4 | 1.9 | 6.1 | 1.5 | 4.2 | 1.0 | 2.9 |
| Rothamsted | 4.9 | 14 | 4.7 | 14 | 3.7 | 11 | 2.0 | 6.1 | 2.3 | 7.0 |
| Ultuna | 3.2 | 12 | 3.3 | 12 | 0.8 | 2.8 | 0.5 | 1.5 | 0.8 | 2.9 |
| Versailles | 6.6 | 26 | 6.9 | 28 | 7.2 | 29 | 7.5 | 30 | 5.4 | 22 |


**Table 2**: The root mean square error (RMSE) as well as the mean relative error respective to the observation for the three last measurement at each site. The unit for the RMSE values is t C ha$^{-1}$






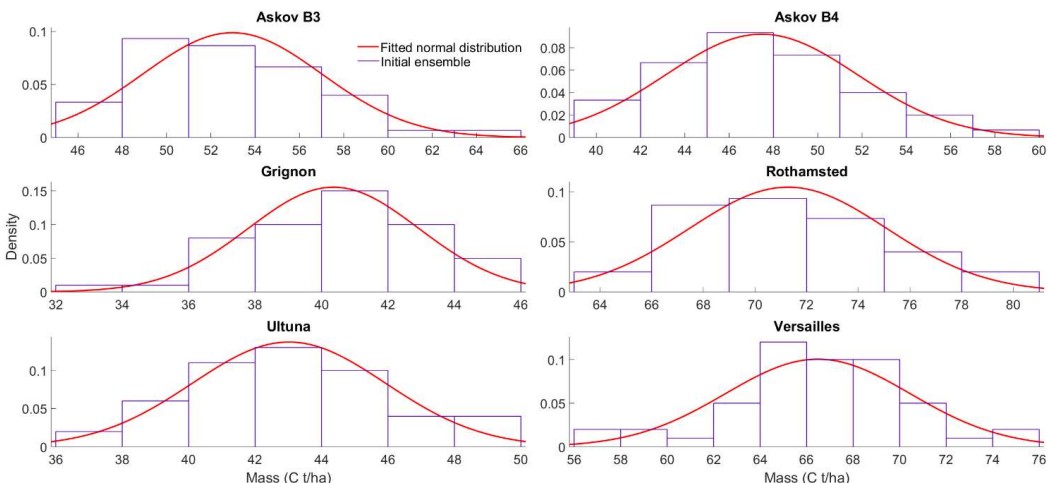

**Figure 1**: The initial ensemble states of total soil carbon (t/ha) at the study sites in the beginning of the fallow campaign.





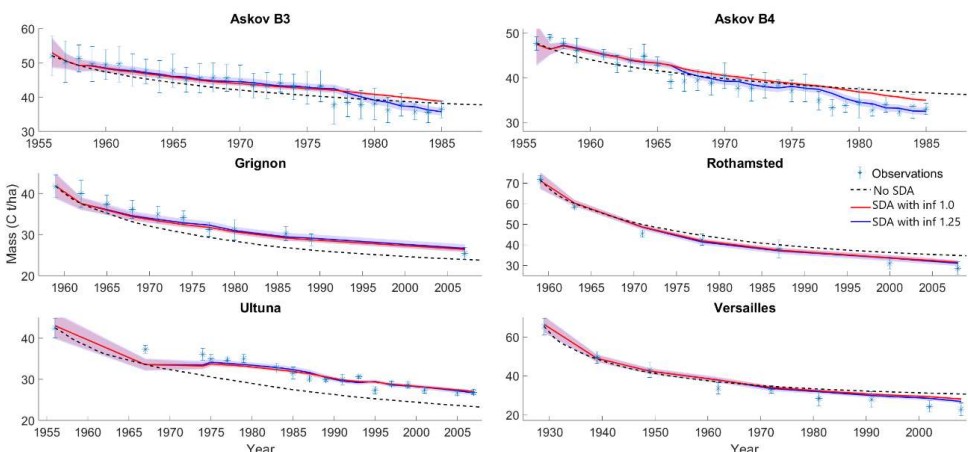

**Figure 2**: Observed and modelled SOC with and without data assimilation (SDA) of all previous measurements using two
different inflation factors (inf). The coloured area around the two different SDA estimates are the 95 % confidence interval.
SDA with higher inflation factor improved predictions at all sites while the, the SDA with lower inflation factor was susceptible
to filter divergence.




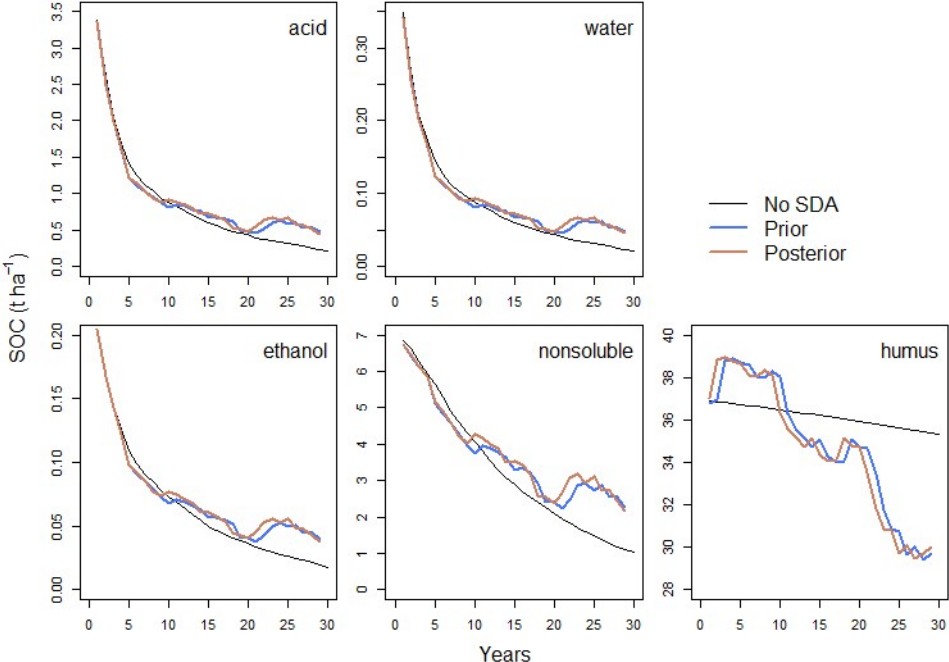

**Figure 3**: The mean of prior and posterior distribution SOC pools at Askov B4 before and after each assimilated observation.
Different SOC pools showed different responses to SDA where humus pool was adjusted the most in response to the
observations and always to their direction. AWEN pool dynamics responded SDA later over the course of assimilation.

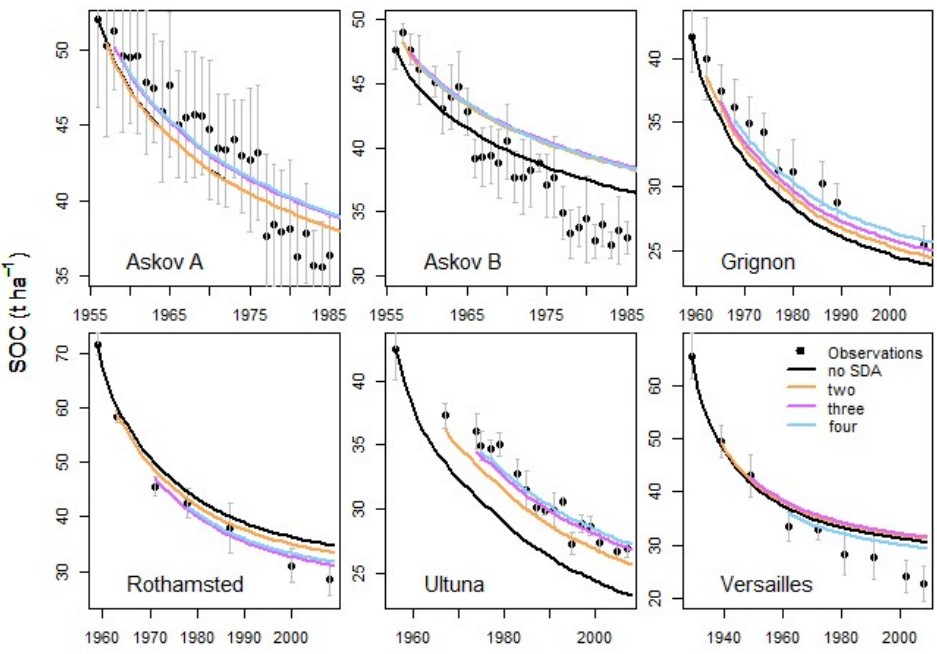

**Figure 4**: Observed and forecasted SOC without data assimilation (black) and with 2-4 initial observations assimilated (coloured lines). Incorporating information already from the first two observations had a noticeable impact on the time series prediction.



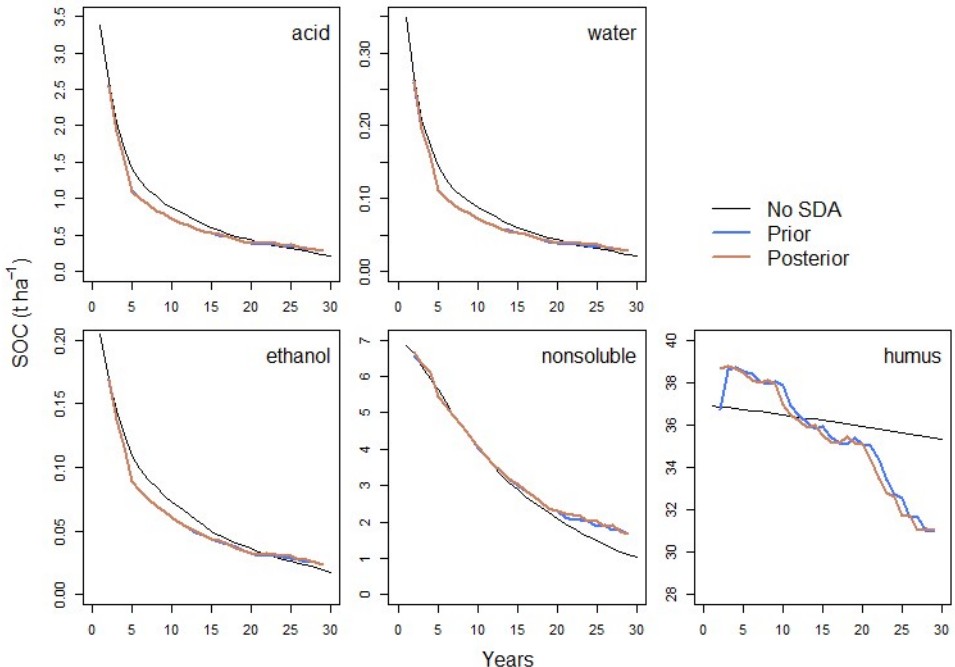

**Figure 5**. The mean prior and posterior distribution SOC pools at Askov B4 before and after each assimilated observation if the assimilation begins from the second observation. While the AWEN pools still show an opposite shift to the H pool later in the assimilation cycle, it is smaller than in Fig 3.
