# Peer review of "Improving Yasso15 soil carbon model estimates with Ensemble Adjustment Kalman Filter state data assimilation"

_Geoscientific Model Development, 2020_

## Referee Comment (RC1) · Anonymous Referee #1 · 16 Jul 2020

The manuscript "Soil carbon estimates by Yasso15 model improved with state data assimilation" by Viskari et al. summarizes the results of incorporating state data assimilation routine (Ensemble Adjustment Kalman filter) into Yasso15 model. The authors demonstrate that assimilating multidecadal observations of soil carbon (C) stocks from bare fallow fields substantially improves model performance. The manuscript aligns well with the aims and scope of Geoscientific Model Development journal, and I would recommend it for publication after minor revisions.

Although given the journal's aims and scope it is not a major issue, the manuscript did not address the mechanistic underpinnings of the observed soil C dynamics in great detail. Posterior soil C dynamics revealed that faster decomposing pools are not in constant decline, and increased after 20 years of no C input. What is the underly-

ing mechanism that could cause this? Is this mechanism accounted for in Yasso15 model structure (e.g. is there C transfer from humus pool to acid-, water-, ethanol-, and nonsoluble pools)? Insights into potential shifts in soil C dynamics may be more easily interpreted when the error between model output and observations is used to also inform model parameters (by evaluating temporal dynamics in the values of the calibrated parameters). However, it is a non-trivial amount of work to incorporate this aspect into the study and analyze the findings, and perhaps should be the focus of another paper.

For the manuscript to be reproducible and more accessible to the reader, I suggest adding a few details. First, it is not completely clear what data was assimilated into the model: was it bulk soil C observations or was it observations of all five soil C pools as in the initial state? These details need to be added to section 2.2. Second, it was not completely clear how the parameters associated with each process and the inputs were estimated. In lines 103-106 authors state that Adaptive Metropolis MCMC (which, unlike EnAKF, is a batch data assimilation technique) to inform model parameters using a suite of observations of litter decay. Was this work done in this study or a prior study? If it's the former, including the results of assimilation of litter decay parameters would be useful, and if it's the latter, then I suggest including a reference to that work. Regarding the pre-agriculture soil C input estimation, were the estimates included in the z vector? Or was soil C input estimated outside of the data assimilation routine? I think including a conceptual figure describing the steps in the analysis as well as writing out the elements in the vectors listed in the equation 1 and 2 would clarify these aspects of the analysis.

Below, please see the detailed list of suggestions:

L45-46: it is not clear to me what message this sentence is trying to communicate. Does it mean that a lot of samples are needed to reliably estimate SOC?

L100: is there a partitioning matrix for soil C input b that distributes it among the five

soil C pools? Also, I'd suggest writing out A explicitly for the reader to understand C transfer pathways between the pools better.

L102-103: I suggest writing out the environmental limitation functions.

L127: a comma is needed between "observation" and "and"

L130: Please provide full matrix A.

L133: Please provide the calculation for Pf

L146-147: it is unclear what "all the associated challenges" are referring to

L275: I suggest removing "here" and adding a comma before "where"

L334: Please specify what should be the focus of the additional study. Also, I think there is a typo: should it be "simple", not "simply"?

---

## Referee Comment (RC2) · Anonymous Referee #2 · 26 Aug 2020

Referee #2 comments:

Review of "Soil carbon estimates by Yasso15 model improved with state data assimilation"by Viskari et al. 2020

General comments:

Viskari et al. applied the Ensemble Adjustment Kalman filter state data assimilation method (SDA) with soil carbon stock time series from six bare fallow experiments to improve the Yasso15 soil carbon model estimates. The application of SDA for the Yasso soil cabron model is unique and showed that the perfomance of the model improved already with few observations while model uncertainties decreased at least on three sites (Rothamsted, Grignon, Ultuna). Less improvement and higher uncertainties were

found on the sites with higher decrease of late measurments than what could be expected by extrapolating early exponential decay trends. The analysis also showed that the estimates of humus pool could have declined more than predicted by default/no-SDA informed model while the more expectedly labile AWEN pools could have lately increased.

These were relevant findings of the study and were discussed mostly in relation to the method. However, the results of this study should be also discussed with similar studies applying Ensemble Kalman filter in the carbon modelling (Eg. Trundiner et al. 2008, Gao et al. 2011, Yan et al. 2019). For example analysis of parameters which was missing in this study in Gao et al. 2011 demonstrated that higher uncertainty of humus pool is correlated with poorly constrained exit rates to this pool.

The text should also be improved for clarity e.g. by separating ideas from lenghty sentences. Especially conclusions need reformulation e.g. large emphasis of conclusions included issues not studied by authors. Overall after the major text revision, the paper could be due to the interesting idea and unique results a valuable contribution to GMD journal and I could recommend it for publication.

Complying with the scope of GMD authors provided relatively clear code and simple tutorial. However, the analysis could not be easily replicated as some parts of the code are missing e.g. Yasso15 model function in R, and e.g instructions of DART analysis (test bed fof data assimilation) were in Finnish language. Please consider revising.

Specific comments:

- Title : Improving Yasso15 soil carbon model estimates with Ensemble Adjustment Kalman filter state data assimilation

- add short description of Ensemble Adjustment Kalman filter to introduction

L24 does the reference apply for peatlands and deforestation?

L44 remove repetition of future uncertainties

L144-154 please reformulate, separate sentences

L173-176 please reformulate, separate sentences

L198-205 clarify by equation or method reference? mention error covariance matrix?

L220 add description of factors 1 and 1.25 included in the Fig. before describing not shown 1.5

L230 Although,...

L234 values (at Askov B4 after 1966, and at askov B3 after 1977).

L240 ... difference. However, ...

Results described in the main text but not shown in main figures could be shown in the supplement?

L255 replace close to each other by similar

L257 not clear, success of SDA forecast depends on inflation factors of error covariance matrix?

L265 delete one

L332 not clear

L335-339 please reformulate, increasing number of assimilated variables was not studies here

L339-341 please reformulate, why to estimate SOC in forest for agricultural C management

- check GMD format of references, add DOI numbers to references when available

References mentioned above:

Gao, C., Wang, H., Weng, E., Lakshmivarahan, S., Zhang, Y., & Luo, Y.: Assimilation of

multiple data sets with the ensemble Kalman filter to improve forecasts of forest carbon dynamics. Ecological Applications: A Publication of the Ecological Society of America, 21(5), 1461–1473, https://doi.org/10.1890/09-1234.1, 2011.

Trudinger, C. M., Raupach, M. R., Rayner, P. J., & Enting, I. G.: Using the Kalman filter for parameter estimation in biogeochemical models. Environmetrics, 19(8), 849–870, https://doi.org/10.1002/env.910, 2008.

Yan, M., Li, Z., Tian, X., Zhang, L., & Zhou, Y.: Improved simulation of carbon and water fluxes by assimilating multi-layer soil temperature and moisture into process-based biogeochemical model. Forest Ecosystems, 6(1), 12, https://doi.org/10.1186/s40663-019-0171-5, 2019.

———————————————

---

## Author Comment (AC1) · 22 Sep 2020

To begin, we would like to thank both reviewers for their time and effort. All the given feedback and comments were constructive and the suggested revisions increased the clarity for the manuscript. The remainder of this text provides point-to-point responses and details on how we have addressed the comments. The line numbers refer to the lines in the change-tracked version of manuscript.

Response to Reviewer #1:

[AC1]:'Anonymous Referee #1 The manuscript "Soil carbon estimates by Yasso15 model improved with state data assimilation" by Viskari et al. summarizes the results of incorporating state data assimilation routine

(Ensemble Adjustment Kalman filter) into Yasso15 model. The authors demonstrate that assimilating multidecadal observations of soil carbon (C) stocks from bare fallow fields substantially improves model performance. The manuscript aligns well with the aims and scope of Geoscientific Model Development journal, and I would recommend it for publication after minor revisions. Although given the journal's aims and scope it is not a major issue, the manuscript did not address the mechanistic underpinnings of the observed soil C dynamics in great detail. Posterior soil C dynamics revealed that faster decomposing pools are not inconstant decline, and increased after 20 years of no C input. What is the underlying mechanism that could cause this? Is this mechanism accounted for in Yasso15model structure (e.g. is there C transfer from humus pool to acid-, water-, ethanol-,and nonsoluble pools)? Insights into potential shifts in soil C dynamics may be more easily interpreted when the error between model output and observations is used to also inform model parameters (by evaluating temporal dynamics in the values of the calibrated parameters). However, it is a non-trivial amount of work to incorporate this aspect into the study and analyze the findings, and perhaps should be the focus of another paper.'

The point raised by the reviewer concerning the mechanistic underpinnings of the Yasso model is exact and we have been interested to work it out ourselves at some point. We agree, though, that it is beyond the scope the current work here. However, in the case referred to here where the analysis increased the AWEN pool concentrations late into the decomposition process was the method reacting to an unexpected change rather than what was likely actually happening in the system at that point. We expanded the discussion paragraph related to the matter, starting from line 356, to be make this more explicit:

"In addition to providing a valuable illustration in how the error covariances change over time and impact the later state corrections, the Askov sites also highlight both a particular strength and limitation of the SDA methods. As seen in the measurement time series (Fig 3), both Askov B3 and B4 have a systematic shift in measurements at

different times due to reasons currently not known. The model projections alone cannot capture these developments as it is unclear if the sudden drop in observed SOC is even due to an ecological process not represented in the model or some issues relating to the measurements. In either case, the inflated SDA adapts to the new state within a few measurement cycles and produces a forecast that follows the new state well. This is clearly a strength of the SDA method that would be beneficial when forecasting SOC at locations where there are disturbances and alterations in the surface conditions. However, here the new state estimate appears questionable as there is a sudden increase in active SOC (i.e. AWEN pools) despite it being over a decade since there was any litter fall. This goes against the current understanding of the system as there is no transference from the H pool back to the AWEN pools and no litter to provide those faster decomposing carbon compounds. In this case, the late increase of the AWEN pool concentrations at the Askov sites are an artefact of the error covariances established earlier rather than a realistic representation of what is happening in the soil. Thus, while SDA is a beneficial tool when examining changing systems, the nature of SOC model dynamics makes it important to also expertly assess how the new estimated state reacts to those changes."

[AC1]:'For the manuscript to be reproducible and more accessible to the reader, I suggest adding a few details. First, it is not completely clear what data was assimilated into the model: was it bulk soil C observations or was it observations of all five soil C pools as in the initial state? These details need to be added to section 2.2.'

As suggested by the reviewer, we added to section 2.2 a clarification that the measurements are for the bulk soil C and that there are no fractioned AWEN measurements for this time series. The added sentence is on line 134: "All the SOC measurements are of the bulk soil C without details on AWEN fractions."

[AC1]:'Second, it was not completely clear how the parameters associated with each process and the inputs were estimated. In lines 103-106 authors state that Adaptive Metropolis MCMC (which, unlike EnAKF, is a batch data assimilation technique) to

inform model parameters using a suite of observations of litter decay. Was this work done in this study or a prior study? If it's the former, including the results of assimilation of litter decay parameters would be useful, and if it's the latter, then I suggest including a reference to that work. '

The Yasso15 calibration has not been published as an independent study, but it has been used in previously published research such as in 'Comparing soil inventory with modelling: Carbon balance in central European forest soils varies among forest types' by Ziche et al. (2019) in Science of the Total Environment. As a response to the request here, we included a supplemental table with both the estimated parameter values for the equations shown and uncertainty ranges for them.

[AC1]: 'Regarding the pre-agriculture soil C input estimation, were the estimates included in the z vector? Or was soil C input estimated outside of the data assimilation routine?'

The pre-agricultural SOC estimation was done before the assimilation cycle and the initial state for the SDA process is calculated by first approximating the SOC before the start of the cultivation and then how it changes after the cultivation starts. We moved the description on the steps before the period of bare fallow from 2.4. to 2.2.

[AC1]: 'I think including a conceptual figure describing the steps in the analysis as well as writing out the elements in the vectors listed in the equation 1 and 2 would clarify these aspects of the analysis.'

We added the conceptual figure and expanded section 2.2 to more clearly explain how the initial state was determined. The section now reads as follows starting from line 129:

'Bare fallow experiments included in the study were kept vegetation-free and free of organic amendments for more than 25 years. The study sites are located in Europe and selected characteristic of these are presented in Table 1. The cultivation time that

lead-up to the bare fallow experiment varied from 75 years to centuries. All the SOC measurements are of the bulk soil C without details on AWEN fractions. The sites are introduced in detail by Barré et al. (2010) and Menichetti et al (2019).

Yasso model requires information on the initial AWENH fractions which we estimated site-specifically based on the Net Primary Production (NPP) in the presumable native habitat and the estimated litter input during time of cultivation (Figure 1, Table 1, Kulmala and Liski, 2018). In short, we first made a general NPP estimate in the native habitat using mean temperature and precipitation as in Del Grosso et al. (2008) and divided it into non-woody, small-sized woody and large woody litter fractions based on the native ecosystem. The different litter types had individual carbon fractions based on solubility. Next, we used Yasso to determine the steady state pool of soil carbon and its fractions using the NPP, different litter fractions, their chemical fractions and mean temperature and precipitation as driver data (Fig1-a).

Before bare fallow, each field had been cultivated for 75–300 years. For that period, we simulated SOC starting from the steady state SOC and its chemical fractions achieved using the pre-agriculture litter approximating (Fig1-b), The same mean annual temperature and precipitation were used as drivers for both the pre-agriculture and agriculture SOC decomposition. The annual litter input for the cultivation period was estimated in a site-specific manner to meet the first SOC measurement after the cultivation period. The carbon fractions in that litter input were assumed to be as presented in Karhu et al. (2012) and the litter is assumed to be non-woody with a diameter of 0 cm. The resulting SOC as the starting point for the bare fallow period (Fig1-c). The AWENH distributions calculated in this manner for each site are shown in Table 1 and are used to calculate the zf from eq. 4 for the first assimilation cycle as detailed below.'

[AC1]: 'L45-46: It is not clear to me what message this sentence is trying to communicate. Does it mean that a lot of samples are needed to reliably estimate SOC?'

We thank the reviewer for pointing this out. We were attempting to argue here that

because measurements are so difficult, limited amount of measurements are often used to represent wider regions which will inherently introduce error due to high spatial SOC variability. Here we split the sentence in two separate senteces to better explain this as follows starting from line 47: 'Relevant measurement campaigns are resource-heavy and time-costly (Jandl et al., 2014). Consequently, single measurements are used to represent SOC concentrations for wider regions despite SOC varying highly spatially, which will inherently introduce error into SOC projections.'

[AC1]: 'L100: Is there a portioning matrix for soil C input b that distributes it among the five soil C pools? Also, I'd suggest writing out A explicitly for the reader to understand C transfer pathways between the pools better.'

The soil C input b is divided into the same chemical composition defined pools as the Yasso state vector and thus there is no need for a portioning matrix. The division is determined by the litter species. This explanation was added to line 110 that reads: 'Where xt is a vector where each component is the amount of carbon in each pool and the litter input b is divided into the same chemical pool components as the Yasso SOC pools which added directly to the YASSO state vector. The division of b depends on the litter species.'

We also added the explicit equation for matrix A starting from line 113 but for clarity, we call it M in the revised manuscript (motivated later).

[AC1]: 'L102-103: I suggest writing out the environmental limitation functions.'

Wrote the equation out as equation 3 on line 119.

[AC1]: 'L 127: a comma is needed between "Observation" and "and"'

Added the comma

[AC1]: 'L130: Please provide full matrix A'

We found a mistake on our part here as the matrix A here is different than the one in

equation 1, so we renamed that matrix to M as mentioned above. However, providing the full matrix A is difficult as there is not set matrix, it's calculated separately for each analysis step. Even the original EAKF article referenced here only proves that matrix A exists, but also states that there are multiple matrix As. So there's no full matrix to provide here, especially since this is the general method used here.

[AC1]: 'L133: Please provide the calculation for Pf'

We added an explanation and equation starting from Line 178: "In Ensemble Kalman Filter applications, each component Pfij of the forecast error covariance matrix Pf is calculated over the ensemble members $P\_ij\hat{}f = (\sum\_{(k} =$ $1) L(z\_{(i, k)}\hat{}f - z\_\hat{}f)(z\_{(j, k)}\hat{}f - z\_\hat{}f))/(L - 1)(7.) Where L is the size of the ensemble."$

[AC1]: 'L146-147: it is unclear what "all the associated challenges are referring to."'

It was referring to the previously explained uncertainty sources, but it was admittedly a confusing statement. We removed that part of the sentence from the manuscript as the following part of the paragraph details what filter divergence is and rephrased the segment in general for clarity. The new paragraph, starting from line 185, reads:

"There are practical challenges that need to be accounted for when utilizing SDA methods, filter divergence (Schlee et al, 1967) being the most relevant one regarding this study. In practice, the ensemble uncertainty does not represent all the uncertainty sources affecting the model predictions. For example, process error arises due to underrepresented model processes or in-sufficiently included model interactions. As there are no reliable ways to establish it for process-based simulators like YASSO (van Oijen, 2017), it cannot be accounted for in the assimilation process. Consequently, the modelled uncertainty does not necessarily increase enough to balance out the reduction in posterior uncertainty during the analysis phase. As a result, the updating process gives too much weight to the prior state in comparison to the observed state as the projected uncertainty decreases while the observation uncertainty remains similar. Ultimately a stage is reached where the measurements stop affecting the estimate due to

the difference in uncertainties. When this happens, the forecast begins to diverge from reality. There are several methods for dealing with filter divergence (Evensen, 2004; Anderson, 2006), but as this a preliminary study, we used a simple inflation method established in Hamill (2001). In this approach, the forecast/prior covariance is multiplied with a constant factor greater than 1 before every analysis/update step to ensure that the measurement continues to affect the estimate. The practical implementation is explained in more detail in the following section."

[AC1]: 'L275: I suggest removing "here" and adding a comma before "where"'

Both corrections done in the manuscript.

[AC1]: 'L334: Please specify what should be the focus of the additional study. Also, I think there is a typo: should be simple, not simply?'

We added how to address the filter divergence as an example of additional study. We also corrected the typo, thank you for pointing that out. The start of the Conclusions from line 412: "The work also highlights the need for additional study such as, for example, how to best address the filter divergence issue or what is driving the differences in how SDA performs at different sites." It should be pointed out that in response to Reviewer 2's comments, the Conclusions section has been rewritten to put more focus on the results from here.

  Response to reviewer #2:

[AC2]: 'Referee #2 comments: Review of "Soil carbon estimates by Yasso15 model improved with state data assimilation" by Viskari et al. 2020

General comments: Viskari et al. applied the Ensemble Adjustment Kalman filter state data assimilation method (SDA) with soil carbon stock time series from six bare fallow experiments to improve the Yasso15 soil carbon model estimates. The application of SDA for the Yasso soil cabron model is unique and showed that the perfomance of the model improved already with few observations while model uncertainties decreased

at least on three sites (Rothamsted, Grignon, Ultuna). Less improvement and higher uncertainties were found on the sites with higher decrease of late measurments than what could be expected by extrapolating early exponential decay trends. The analysis also showed that the estimates of humus pool could have declined more than predicted by default/no-SDA informed model while the more expectedly labile AWEN pools could have lately increased. These were relevant findings of the study and were discussed mostly in relation to the method. However, the results of this study should be also discussed with similar studies applying Ensemble Kalman filter in the carbon modelling (Eg. Trundiner et al. 2008, Gao et al. 2011, Yan et al. 2019). For example analysis of parameters which was missing in this study in Gao et al. 2011 demonstrated that higher uncertainty of humus pool is correlated with poorly constrained exit rates to this pool.'

Thank you for helpfully pointing out these articles. We added a reference to the Gao et al. (2011) paper in the discussion when bringing up how uncertainty in model processes can still limit how much or how fast the SDA can improve future projections. Starting from line 340, the manuscript now reads:

"This site highlights that while SDA is a valuable tool for improving forecasts, it is still limited by how well the applied model captures the local SOC dynamics. For instance, if the soil respiration parameters for the H pool is poorly represented for that location, then the resulting H pool, as well as the total SOC, projections would still be more uncertain even when applying SDA methods (Gao et al., 2011)."

In addition, we added a reference to the Yan paper in discussion starting from line 308: "This study establishes that state data assimilation (SDA) improves soil organic carbon (SOC) forecasts by continuously incorporating total carbon measurements. As such, this adds another type of measurements that SDA can use to improve future projections in addition to previously shown positive impacts of assimilating soil environmental conditions (Yan et al., 2019)."

However, including the Trudinger paper to the discussion proved challenging as it touched on theoretically proving that SDA methods can be used to constrain model parameters in these systems, which does not directly relate to the work done here. Instead we acknowledged the previous research in the introduction as a way to illustrate the prior work in SOC systems with SDA methods. Starting from line 65:

"In SOC related systems, SDA has been limited so far and either focused on estimating model parameters (Trudinger et al., 2008) or constraining the drivers affecting the soil carbon fluxes (Yan et al., 2019). In Gao et al., (2011), SOC was estimated as a component of the total carbon allocated in a forest ecosystem and the results showed potential, but even there the main focus was on the model parameter estimation."

[AC2]: 'The text should also be improved for clarity e.g. by separating ideas from lenghty sentences.'

We read through the manuscript in order to identify and change sentences that would could be improved for clarity.

[AC2]: 'Especially conclusions need reformulation e.g. large emphasis of conclusions included issues not studied by authors'

The Conclusions section has been revised to highlight more the work done here. There is still some discussions of studies not done here, but they are more clearly framed as matters regarding the implementation of these methods which need to be studied going forward. The Conclusions section, starting from line 411, now reads:

"The results here show that there are benefits in implementing SDA methods in SOC research as assimilating the first few observations was generally sufficient in improving long-term SOC projections when compared to later measurements. Furthermore, the SDA methods here successfully used coarse bulk C measurements to update the more detailed model state with the developing error covariance matrices connecting the different state variables. The work also highlights the need for additional study such as,

for example, how to best address the filter divergence issue or what is driving the differences in how SDA performs at different sites. The focus here was in a very simple system where there was no litter input and on a specific SDA method with its own benefits and hindrances. Increasing the complexity of the system, such as by introducing different types of litter, using measurements from other locations to estimate local SOC or incorporating flux tower respiration measurements to constrain projected SOC changes, will raise new practical challenges that have to be addressed in future work. Still, by allowing actively incorporating multiple information sources, SDA is a crucial tool for all process-based model projections, for example approximating the amount of SOC in a forest or assessing how agricultural carbon allocation changes in response to field management."

[AC2]: 'Overall after the major text revision, the paper could be due to the interesting idea and unique results a valuable contribution to GMD journal and I could recommend it for publication. Complying with the scope of GMD authors provided relatively clear code and simple tutorial. However, the analysis could not be easily replicated as some parts of the code are missing e.g. Yasso15 model function in R, and e.g instructions of DART analysis (test bed of data assimilation) were in Finnish language. Please consider revising.'

Our deepest gratitude for noticing the error in language in the DART analysis instructions, we included the wrong file accidentally in the package. We have added the requested files, as well as the example DART files, now into the uploaded package to make repeating the analysis easier.

Response to specific comments:

[AC2]: 'Title : Improving Yasso15 soil carbon model estimates with Ensemble Adjustment Kalman filter state data assimilation'

Title has been changed as suggested.

[Figure]

[AC2]: 'add short description of Ensemble Adjustment Kalman filter to introduction'

Added a short description to the end of the last paragraph of the introduction, starting from line 89. The new part reads "We applied the Ensemble Adjustment Kalman filter (EAKF; Andersson, 2001) as the SDA method in the study. In EAKF, the ensemble is created by running the model with varying initial states, which are then all updated with the information from measurements as explained in more details in the following sections. Not only is EAKF a widely established SDA method, but it is a part of the Data Assimilation Research Testbed (DART; Anderson et al, 2009) workflow."

[AC2]: 'L24 does the reference apply for peatlands and deforestation?'

Yes, for peatlands and deforested areas the vegetative input of carbon through roots, litterfall and forest harvest residues are still relevant even though there is no substantial surface vegetation remains. For the sake of clarity, on line 25 we changed it from 'surface vegetation litter' to simply 'vegetation litter'.

[AC2]: 'L44 remove repetition of future uncertainties*

We removed the latter use of the term.

[AC2]: 'L144-154 please reformulate, separate sentences'

Reformulated the paragraph so that it now focuses more clearly on the filter divergence issue relevant here. The new paragraph, starting from line 185, is below.

"There are practical challenges that need to be accounted for when utilizing SDA methods, filter divergence (Schlee et al, 1967) being the most relevant one regarding this study. In practice, the ensemble uncertainty does not represent all the uncertainty sources affecting the model predictions. For example, process error arises due to underrepresented model processes or in-sufficiently included model interactions.As there are no reliable ways to establish it for process-based simulators like YASSO (van Oijen, 2017), it cannot be accounted for in the assimilation process. Consequently, the modelled uncertainty does not necessarily increase enough to balance out the reduction

in posterior uncertainty during the analysis phase. As a result, the updating process gives too much weight to the prior state in comparison to the observed state as the projected uncertainty decreases while the observation uncertainty remains similar. Ultimately a stage is reached where the measurements stopaffecting the estimate due to the difference in uncertainties. When this happens, the forecast begins to diverge from reality. There are several methods for dealing with filter divergence (Evensen, 2004; Anderson, 2006), but as this a preliminary study, we used a simple inflation method established in Hamill (2001). In this approach, the forecast/prior covariance is multiplied with a constant factor greater than 1 before every analysis/update step to ensure that the measurement continues to affect the estimate. The practical implementation is explained in more detail in the following section."

[AC2]: 'L173-176 please reformulate, separate sentences'

We reformulated the sentence referred to here, the current version starting from line 224 reads

"A 10 % mean was used as the initial ensemble variance as there were no reliable initial state uncertainty approximation. This was found to be larger than just perturbing the litter inputs and, as such, was not likely to underestimate the initial uncertainty."

[AC2]: 'L198-205 clarify by equation or method reference? mention error covariance matrix?'

We added the references to eqs. 4-6 to the paragraph. We also explained that the inflation factor was used to scale the posterior error covariance matrix produced by eq. 6 and that is how it affects the state estimates produced by eqs. 4 and 5. The revised version starts from line 256 and reads:

"The SDA was first tested by updating the model state variables at each measurement time according to eq. (4). The produced state estimation was then used by Yasso to determine the next predicted state. This basic test was repeated with three different

inflation factors (1, 1.25 and 1.5) in order to examine how much filter divergence affects the projections and which inflation factor range produces satisfactory predictions. The inflation factors were implemented by scaling the posterior error covariance matrix produced by eq. 6 according to the inflation factor chosen before using it in eq. 5, and consequently, in eq. 4."

[AC2]: 'L220 add description of factors 1 and 1.25 included in the Fig. before describing not shown 1.5'

The uninflated and inflated SDA predictions refer to the inflation factor 1. and 1.25 runs in Fig 2. We clarified this by adding inflation factor 1 and inflation factor 1.25 in paragraphs when first referring to the uninflated and inflated SDA predictions.

[AC2]: 'L230 Although,...'

Separated the two sentences. Also shifted the focus with the although part only to the Versailles as the lack of change is much more visible with it than with Rothamsted.

[AC2]: 'L234 values (at Askov B4 after 1966, and at askov B3 after 1977).'

Changed the text as suggested.

[AC2]: 'L240 ... difference. However, ...' We did not know how to separate this sentence as suggested as the next part is what the difference noted was. To clarify we changed the ',however,' where' to 'as'.

[AC2]: 'Results described in the main text but not shown in main figures could be shown in the supplement?'

Added the 1.5 inflation factor assimilation figure and the AWENH pool figures for the other sites as supplemental figures. We did not add the comparison figures for the runs where the first observation was not included as with the exception of the AWENH pool figures for the Askov sites that they were so visually similar to the main result figures that there was no real additional information there, especially considering the amount

of new figures added.

[AC2]: 'L255 replace close to each other by similar'

Changed.

[AC2]: 'L257 not clear, success of SDA forecast depends on inflation factors of error covariance matrix?'

To clarify this, we added a sentence to line 321: "This is expected as the inflation reduces the impact of filter divergence and thus allows for the later observations affect the analysis more than they would without the uncertainty inflation."

[AC2]: 'L265 delete one'

Deleted.

[AC2]: 'L332 not clear'

Expanded the start of the conclusion to be more explicit on the success of the SDA approach as follows starting from line 411: 'The results here show that there are benefits in implementing SDA methods in SOC research as they generally produced SOC projections that fitted better with the later observations than the non-SDA projections. The work here also highlight. . .'

[AC2]: 'L335-339 please reformulate, increasing number of assimilated variables was not studied here'

We rewrote this part of the Conclusions section as detailed in the response to the general comments.

[AC2]: 'L339-341 please reformulate, why to estimate SOC in forest for agricultural C management'

Reformulated to remove the confusing part as forest SOC does not affect agricultural C management. They were meant to be two examples where SDA would benefit SOC

projections and the text has been changed to reflect that.

[Figure]

[Figure]

**Fig. 1.** Conceptual figure of how the initial state for each plot is determined

---

## Author Response (AR2)

Dear editor,

Before anything else, we are grateful that our previous revisions were sufficient and that our manuscript was accepted with minor revisions.

Upon the requested revisions, we changed the link to the Github depository on line 407, apologies on the old link remaining in the manuscript. We also asked a person unaffiliated with the manuscript to read through it and point out parts where the text could be smoother. Those changes have been done in the manuscript.

Hopefully these revisions are considered suitable and the manuscript can move to the final phase.

Toni Viskari
Senior researcher
Finnish Meterological Institute
toni.viskari@fmi.fi

[revised manuscript text omitted]